# Agent-to-Sim: Learning Interactive Behavior from Casual Videos

## Abstract

Agent behavior simulation empowers robotics, gaming, movies, and VR applications, but building such simulators often requires laborious effort of manually crafting the agent's decision process and motion patterns. Recent advances in visual tracking and motion capture have enabled learning agent behavior from real-world data, but these methods are limited to a few scenarios due to the dependence on specialized sensors (e.g., synchronized multi-camera systems). In a step towards scalable and realistic behavior simulators, we present Agent-to-Sim (ATS), a framework for learning simulatable 3D agents in a 3D environment from casually-captured monocular videos. To deal with partial views, our framework fuses observations in a canonical space for both the agent and the scene, resulting in a dense 4D spatiotemporal reconstruction. We then learn an interactive behavior generator by querying paired data of agents' perception and actions from the 4D reconstruction. ATS enables real-to-sim transfer of agents in their familiar environments given longitudinal video recordings captured with a smartphone over a month. We show results on pets (e.g., cat, dog, bunny) and a person, and analyse how the observer's motion and 3D scene affect an agent's behavior.

## 1 Introduction

Consider the scene of the cat in the living room: where will the cat go and how will it move? Since we have seen cats interact with the environment and other people many times, we know that cats like to go to the couch, often move slowly, and follow humans around, but run away if people come too close. Such a predictive model of a physical agent is what enables plausible behavior simulation, which is essential for embodied intelligence, immersive virtual environments and robot planning in safety-critical scenarios [9, 31, 41, 45, 54].

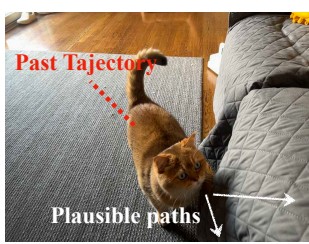

The key challenge with behavior simulation is how to generate *plausible* and *interactive* behavior (with respect to the scene and other agents). On one hand, prior works [2, 6, 46] utilize trajectory computed by path-planning algorithms or hand-designed logic from game simulators [13, 58]. While these approaches benefit from high-quality trajectory data paired with perfect object and scene geometries, it is laborious to manually craft simulators that suit the needs of each type of application, and the data distribution is fundamentally different from the real world, leading to unnatural motion and interactions. On the other hand, vision-based motion capture enables learning plausible behavior directly from data for certain scenarios, such as autonomous driving [9], human body motion [21, 36], and interaction with objects/scenes [14, 24]. However, due to the dependence on specialized sensor (synchronized multi-camera systems, IMUs, pre-scanned objects), such systems does not scale well to the full spectrum of natural behavior one may care about, such as behavior of animals, casual events, and long-term activities.

Submitted to 38th Conference on Neural Information Processing Systems (NeurIPS 2024). Do not distribute.

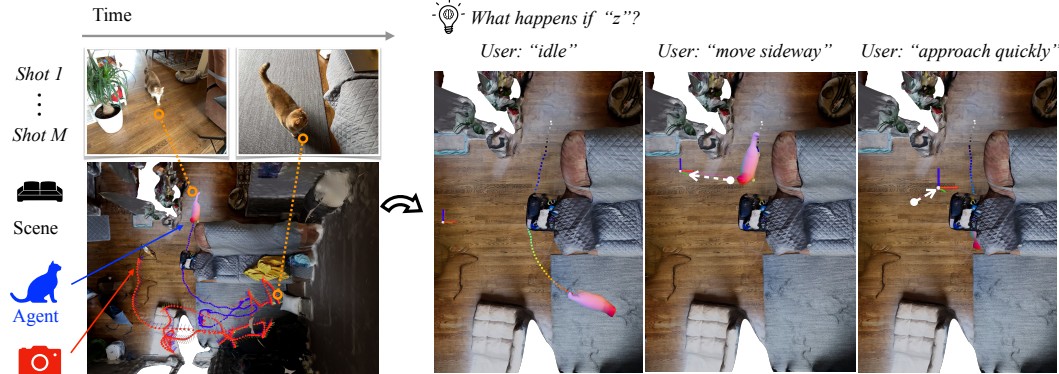

Figure 1: **Learning agent behavior from longitudinal casual video recordings.** We answer the following question: can we simulate the behavior of an agent, by learning from casually-captured videos of the *same* agent recorded across a long period of time (*e.g.*, a month)? A) We first reconstruct videos in 4D (3D & time), which includes the scene, the trajectory of the agent, and the trajectory of the observer (i.e., camera held by observer). Such individual 4D reconstruction are registered across time, resulting in a *complete* 4D reconstructions. B) Then we learn a representation of the agent that allows for interactive behavior simulation. The behavior model explicitly reasons about goals, paths, and full body movements conditioned on the agent's ego-perception and past trajectory. Such agent representation allows us to simulate novel scenarios through conditioning. For example, conditioned different observer trajectories, the cat agent choose to walk to the carpet, stays still while quivering his tail, or hide under the tray stand. *Please see videos and results of other agents in the supplement.*

Recent advances in differentiable rendering [10, 12, 23, 38, 42, 52, 59, 65] and monocular MoCap [28, 43, 69, 70] provide a pathway to obtain high-quality models of scenes and agents from monocular videos alone. Despite the potential of covering diverse data of agent behavior that match the real-world distributions, none of the existing works brings a solution of reconstructing dense 3D structures of both the agent and scene, which is crucial for learning agent behavior grounded in real world environments. To address this, we present ATS (Agent-to-Sim), a framework for learning simulatable agent from casual videos captured over a long time horizon (*e.g.* 1 month), as shown in Fig. 1.

The crucial technical challenge is the presence of partial visibility – in each video captured from an observer's viewpoint, only parts of the agent and the environment are visible. *How do we infer the states of agent and the environment that are not visible?* To build a dense 4D spatiotemporal reconstruction, our key insight is to leverage the observations from multiple videos by fusing them in a canonical 3D space. We introduce a novel coarse-to-fine registration approach that re-purposes "foundational" visual features [40] as a neural localizer, which "registers" the camera with respect to a canonical structure. This enables capturing interactive behavior data in a casual setup (*e.g.*, with a smartphone), and provides paired training data of perception and action of an agent that is grounded in a natural environment (Fig. 2). To learn an interactive behavior model, we condition the action of an agent on their ego-perception, and leverage diffusion models [18, 53] to account for the multimodal nature of goals and planned trajectories. The resulting framework, ATS, can simulate interactive behaviors like those described at the start: agents like pets that leap onto furniture, dart quickly across the room, timidly approach nearby users, and run away if approached too quickly. Our contributions are summerized as follows:

1. **Agent-to-Sim (ATS) Framework.** We introduce a real-to-sim framework, ATS, to learn simulators of interactive agent behavior from casually-captured videos. ATS learns plausible agent behavior that matches the real-world, and is scalable to diverse scenarios, such as animal behavior and casual events.

2. **Environment-Interactive Behavior Simulation.** ATS learns behavior that is *interactive* to the environment, including both the observer and 3D scene. We show the first result of generating plausible behavior of animals that are reactive to observer's motion, and are aware of the 3D scene.

Table 1: **Related works** in behavior data capture. ATS is the only method that builds a complete 4D reconstruction of both the agents and the environment. Different from prior work that focus on specific domains, ATS can be applied to capture interactive behavior of both animals and humans from casual RGBD videos (*e.g.* captured by a smartphone).

| Method | Agent Model | Scene Model | Capture Setup | Domain |
| --- | --- | --- | --- | --- |
| UCY [30] & ETH [44] | Point | N.A. | Manual Anno. | Pedestrian |
| nuScenes [9] | Point | Dense 3D Map | Manual Anno. | Pedestrian, Vehicle |
| SAMP [14] | Parametric Body | Furniture & Objects | Multi-Camera | Human |
| AMASS [36] | Parametric Body | N.A. | Multi-Camera | Human |
| ActionMap [47] | Action Class | Sparse 3D Map | Egocentric Camera | Human |
| ATS (Ours) | Non-parametric | Dense 3D Map | Casual RGBD | Animal, Human |

3. **Complete 4D Registration & Reconstruction.** We present a method to register and reconstruct a temporally-evolving 3D scene, whiling accounts for changes in scene layout and appearance.

## 2 Related Works

**Behavior Prediction and Generation.** Behavior prediction has a long history, starting from simple physics-based models such as social forces [17] to more sophisticated "planning-based" models that cast prediction as reward optimization [26, 76], where the reward is learned via inverse reinforcement learning [75]. With the advent of large-scale pedestrian and vehicle motion data collected in the navigation and autonomous driving domains [1, 34, 37, 48, 50], generative prediction models such as diffusion models have been able to express behavior multi-modality while being easily controlled via additional signals such as cost functions [20] or logical formulae [74]. However, to capture plausible behavior of agents, these approaches are extremely dependant on high-quality agent trajectory data collected "in the wild" with the associated scene context (*e.g.*, 3D map of the scene) [9]. Such data are often manually annotated at a bounding box level (Tab. 1), which limits the scale and the level of detail they can capture. Beyond autonomous driving setup, existing works for human motion prediction and generation [46, 57, 62] have been primarily using simulated data [6] or motion capture data collected with multiple synchronized cameras [14, 24, 36]. Such data provide high-quality full body motion of human using parametric body models [32], but the interactions with the environment are often restricted to a set of pre-defined furnitures and objects [15, 29, 73]. Furthermore, the use of simulated data and motion capture data inherently limits the realism of these behavior generators, since real agents will behave very differently in their familiar environment. To bridge the gap, we develop 4D reconstruction method to obtain high-quality trajectories of agents in their natural environment, with a simple setup that can be achieved with a smartphone. Close to our setup, ActionMap [47] associate daily actions performed by a human agent with an reconstructed 3D environment given egocentric videos. However, they focus on actions performed by hand and do not reconstruct the full body motion of the agent.

**4D Reconstruction from Monocular Videos.** Reconstructing agents and the environment from monocular videos is challenging due to its under-constrained nature. Given a monocular video, there are multiple different interpretations of the underlying 3D geometry, motion, appearance, and lighting [56]. As such, reconstructing agents often require category-specific 3D prior (*e.g.*, 3D humans) [11, 27, 32]. Along this line of work, researchers reconstruct 3D humans aligned to the world coordinate with the help of SLAM and visual odometry [28, 69, 70]. Sitcoms3D [43] reconstructs both the scene and human parameters, while relying on shot changes to determine the scale of the scene. However, the use of parametric body models limits the degrees of freedom they can capture, and makes it difficult to reconstruct agents from arbitrary categories which do not have a pre-built body model, for example, animals. Another line of work avoids using category-specific 3D priors and optimizes the shape and deformation parameters of the agent given richer visual signals (*e.g.*, optical flow and object silhouette) [61, 64, 65], which is shown to work well for a broad range of category including human, animals, and vehicles. TotalRecon [52] further incorporates the background scene into the model-free reconstruction pipeline, such that the agent's motion can be decoupled from the camera motion and aligned to the scene space. However, none of the existing methods can reconstruct both the agent and the scene in high-quality. In practice, individual videos may not contain sufficient

views, leading to inaccurate and incomplete reconstructions. Our method registers both the agent and the environment from multiple videos into a shared space, which leverages large-scale data collection to build a high-quality agent and scene model.

## 3   Approach

We describe a method to learn interactive behavior models given longitudinal video recordings of an agent in the same environment. We first build a spatiotemporal 4D reconstruction, including the agent, the scene, and the observer (Sec. 3.1), which is solved by an optimization involving multi-video registration (Sec. 3.2). We then train an interactive behavior model of the agent that is *interactive* with the surrounding environment, including the scene and the motion of the observer (Sec. 3.3).

### 3.1   4D Representation: Agent, Scene, and Observer

Given multiple monocular videos, our goal is to build a dense spatiotemporal 4D reconstruction of the underlying world, including a deformable agent, a background scene, and a moving observer.

The task is ill-posed due to partial visibility – from an observer's viewpoint, the agent and the environment are only partially visible. To deal with this problem, one principle approach is geometric registration, where structures not visible from one view can be inferred from the other views they appear [51]. We build upon this idea to reconstruct a *complete* spatiotemporal model of an agent and their familiar environment by registering videos captured at different time.

**Problem Setup.** Specifically, given images from $M$ videos represented by color and feature descriptors [40], $\{\mathbf{I}_i, \boldsymbol{\psi}_i\}_{i=\{1,\ldots,M\}}$, our goal is to find a 4D spatiotemporal representation that explains the video, while pixels with the same semantics can be mapped to consistent canonical 3D locations. Our representation factorizes the 4D structure into a static component and a time-varying component.

**Static Representation.** $\mathbf{T} = \{\sigma, \mathbf{c}, \psi\}$. We represent the static component as agent fields and scene fields. Both define densities, colors, and semantic features in a canonical space,

$$(\sigma_s, \mathbf{c}_s, \boldsymbol{\psi}_s) = \mathrm{MLP}_{scene}(\mathbf{X}, \boldsymbol{\beta}_i), \tag{1}$$

$$(\sigma_a, \mathbf{c}_a, \boldsymbol{\psi}_a) = \mathrm{MLP}_{agent}(\mathbf{X}), \tag{2}$$

where $\mathbf{X}$ corresponds to a 3D point. To account for structures that change across videos, we modify the scene fields to take a per-video latent code $\boldsymbol{\beta}_i$ as input, which allows fitting video-specific details.

**Time-varying Representation.** $\mathcal{D} = \{\boldsymbol{\xi}, \mathbf{G}, \mathbf{W}\}$. The time-varying component includes a moving observer, represented by the camera pose $\boldsymbol{\xi}_t \in SE(3)$, and the motion of an agent, represented by a set of rigid bodies, $\{\mathbf{G}_t^b\}_{\{b=1,\ldots,25\}}$, referred to as "bones". Given a time $t$, the canonical space of the agent can be mapped to the camera space by blend-skinning deformation [35, 65],

$$\mathbf{X}_t = \mathbf{G}^a \mathbf{X} = \left(\sum_{b=1}^{B} \mathbf{W}^b \mathbf{G}_t^b\right) \mathbf{X}, \tag{3}$$

which computes the motion of a point by blending the bone transformations (we do so in the dual quaternion space [22, 66] to ensure $\mathbf{G}^a$ is a valid rigid transformation). The skinning weights $\mathbf{W}$ are defined as the probability of a point assigned to each bone.

**Rendering.** To turn the 4D representation into images, we sample rays in the camera space, map them separately to the canonical space of the scene and the agent with $\mathcal{D}$, and query values (e.g., density, color, feature) from corresponding fields of the scene and the agent. The values are then combined before ray integration [39, 52]. Consequently, the rendered pixel values are compared against the observations to update the world representation $\{\mathbf{T}, \mathcal{D}\}$.

**Decoupling Agent Motion from Observer.** $\{\mathbf{G}_t^b\}_{\{b=1,\ldots,25\}}$ defines the motion of an agent with respect to the observer. Given the observer, we compute the motion of the agent in the scene space as,

$$\mathbf{G}_t^{b \to s} = \boldsymbol{\xi}_t^{-1} \mathbf{G}_t^b, \tag{4}$$

where the results of extracted trajectories of the agent is shown in Fig. 2

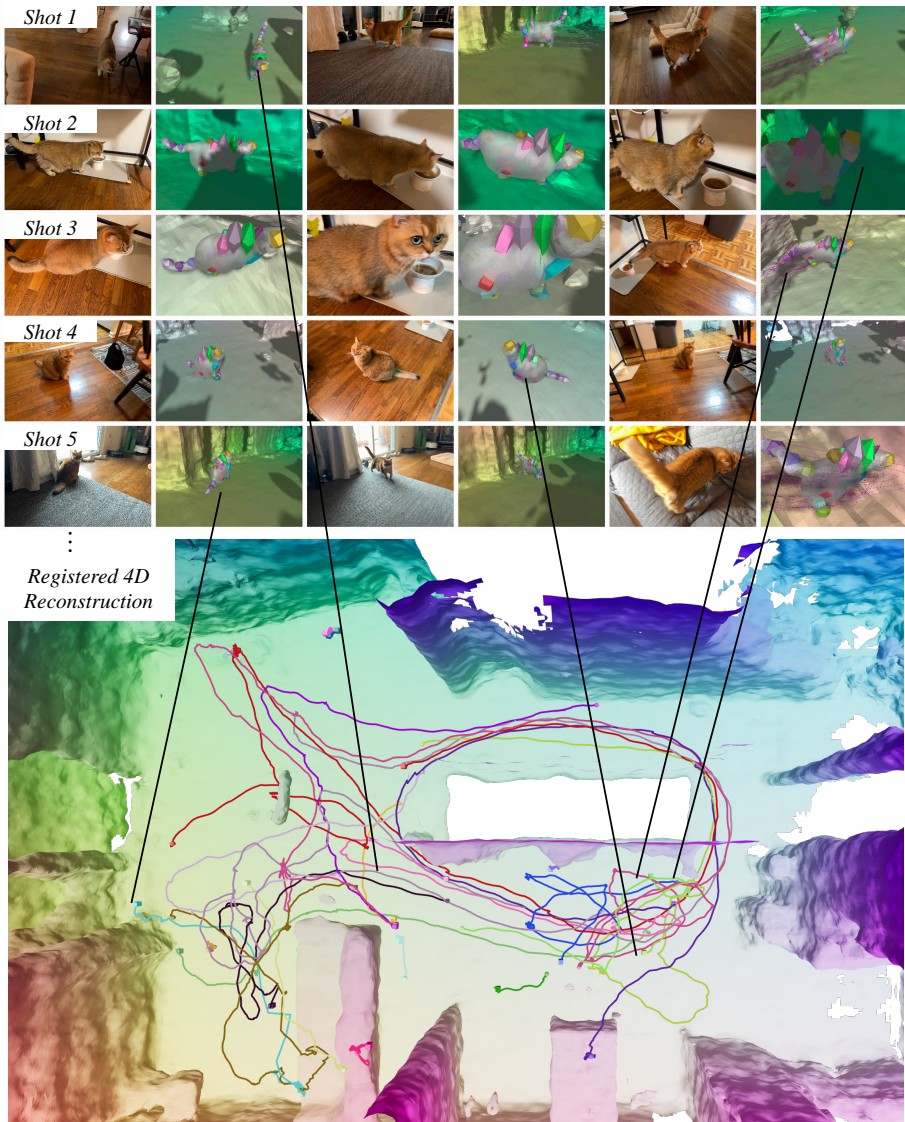

Figure 2: **Results of 4D reconstruction**. Top: reference images and renderings of the reconstructions. The color on the background represents correspondence. The colored blobs on the agent body represent $B = 25$ body parts of the agent (*e.g.*, head is represented by the yellow blob). Bottom: Bird's eye view of the reconstructed scene and agent trajectories, registered to the same scene coordinate. Each colored line represents a unique video sequence where boxes and spheres indicate the starting and the end location. *Please see videos and results on other agents in the supplement.*

## 3.2 Optimization: Multi-Video Registration

To deal with bad local optima caused by camera poses (Fig. 4), we design a coarse-to-fine registration approach that globally aligns the cameras to a shared canonical space with a feed-forward network, and then jointly optimizes the 3D structures while adjusting the cameras locally.

**Initialization: Neural Localization.** Due to the evolving nature of scenes across a long period of time [55], there exist both global layout changes (*e.g.*, furniture get rearranged) and appearance changes (*e.g.*, table cloth gets replaced), making it challenging to find accurate geometric correspondences [4, 5, 49]. With the observation that "foundational" visual features have good 3D and viewpoint awareness [3], we adapt them for camera localization. We learn a scene-specific neural

localizer that directly regresses the camera pose of an image with respect to a canonical structure,

$$\boldsymbol{\xi} = f_\theta(\boldsymbol{\psi}), \tag{5}$$

where $f_\theta$ is a ResNet-18 [16] and $\boldsymbol{\psi}$ is the DINOv2 [40] feature of the input image. We find it to be more robust than geometric correspondence, while being more computationally efficient than performing pairwise matches [49]. To learn the neural localizer, we first capture a walk-through video and build a dense map of the scene. Then we use it to train the neural localizer by randomly sampling camera poses $\mathbf{G}^* = (\mathbf{R}^*, \mathbf{t}^*)$ and rendering images on the fly,

$$\arg\min_\theta \sum_j \left( \| \log(\mathbf{R}_0^T(\theta)\mathbf{R}^*) \| + \|\mathbf{t}_0(\theta) - \mathbf{t}^*\|_2^2 \right), \tag{6}$$

where we use geodesic distance [19] for camera rotation and $L_2$ error for camera translation. For the agent, we follow BANMo [65] to initialize the root pose $\{\mathbf{G}^b\}_{b=0}$ with a pre-trained pose network.

**Objective: Feature-metric Alignemnt.** Given a coarse initialization of the observer (scene camera) and the agent's root pose, we use both photometric and featuremetric losses to optimize $\{\mathbf{T}, \mathcal{D}\}$,

$$\min_{\mathbf{T}, \mathcal{D}} \sum_t \left( \|I_t - \mathcal{R}_I(t; \mathbf{T}, \mathcal{D})\|_2^2 + \|\boldsymbol{\psi}_t - \mathcal{R}_{\boldsymbol{\psi}}(t; \mathbf{T}, \mathcal{D})\|_2^2 \right) + L_{reg}(\mathbf{T}, \mathcal{D}), \tag{7}$$

where $\mathcal{R}(\cdot)$ is the rendering function described in Sec 3.1. In contrast to prior works, using feature-metric errors makes the optimization robust to change of lighting, appearance, and helps find accurate alignment over multiple videos (Fig. 4). The regularization term includes eikonal loss, silhouette loss, flow loss and depth loss similar to prior works [52, 65].

**Scene Annealing.** To encourage the reconstructed scene across videos to share a similar structure, we randomly *swap* the code $\boldsymbol{\beta}$ of two videos during optimization, and gradually decrease the probability of swaps from $\mathcal{P} = 1.0 \rightarrow 0.05$ over the course of optimization. This regularizes the model to effectively share information across all videos, and keeps video-specific details (Fig. 4).

## 3.3 Interactive Behavior Generation

Now that we build a complete 4D reconstruction from multiple videos, we can extract a scene structure $\mathbf{T}$, and $M$ trajectories of the agent $\{\mathbf{G}^t\}_{t=\{T_1,\ldots,T_M\}}$ as well as the observer $\{\boldsymbol{\xi}^t\}_{t=\{T_1,\ldots,T_M\}}$ grounded in the environment. We aim to learn an agent that is interactive with the world.

**Hierarchical Behavior Representation.** We model the behavior of an agent by bone transformations in the scene space $\mathbf{G} \in \mathbb{R}^{6B \times T^*}$ over a fixed time horizon $T^* = 5.6\text{s}$, . We design a hierarchical model as shown in Fig. 3. The body motion $\mathbf{G}$ is conditioned on path $\mathbf{P} \in \mathbb{R}^{3 \times T^*}$, which is further conditioned on goal $\mathbf{Z} \in \mathbb{R}^3$. Such decomposition allows agents to react by predicting goals with low latency

**Goal Generation.** We represent a multi-modal distribution of goals $\mathbf{Z} \in \mathbb{R}^3$ by its score function $s(\mathbf{Z}, \sigma) \in \mathbb{R}^3$ [18, 53]. The score function is implemented as a coordinate MLP [38],

$$s(\mathbf{Z}; \sigma) = \text{MLP}_{\theta_{\mathbf{Z}}}(\mathbf{Z}, \sigma), \tag{8}$$

trained by predicting the amount of noise $\boldsymbol{\epsilon}$ added to the clean goal, given the corrupted goal $\mathbf{Z} + \boldsymbol{\epsilon}$:

$$\arg\min_{\theta_{\mathbf{Z}}} \mathbb{E}_{\mathbf{Z}} \mathbb{E}_{\sigma \sim q(\sigma)} \mathbb{E}_{\boldsymbol{\epsilon} \sim \mathcal{N}(\mathbf{0}, \sigma^2 \mathbf{I})} \|\text{MLP}_{\theta_{\mathbf{Z}}}(\mathbf{Z} + \boldsymbol{\epsilon}; \sigma) - \boldsymbol{\epsilon}\|_2^2. \tag{9}$$

Compared to methods directly learning the multi-modal distribution [8, 25], diffusion models are easy to train and can be used to generate diverse and high-quality samples [18, 53].

**Path Generation with Control.** To guide path generation with goals, we represent its score as

$$s(\mathbf{P}; \sigma) = \text{ControlUNet}_{\theta_{\mathbf{P}}}(\mathbf{P}, \mathbf{Z}, \sigma), \tag{10}$$

where the Control UNet contains two standard UNets with the same architecture [72], one performing unconditional generation taking $(\mathbf{P}, \sigma)$ as input, another injecting goal conditions densely into the neural network blocks of the first one taking $(\mathbf{Z}, \sigma)$ as inputs. Compared to concatenating the goal condition to the noise latent, this encourages close alignment between the goal and the path [62]. We apply the same architecture to control pose generation with paths,

$$s(\mathbf{G}; \sigma) = \text{ControlUNet}_{\theta_{\mathbf{G}}}(\mathbf{G}, \mathbf{P}, \sigma). \tag{11}$$

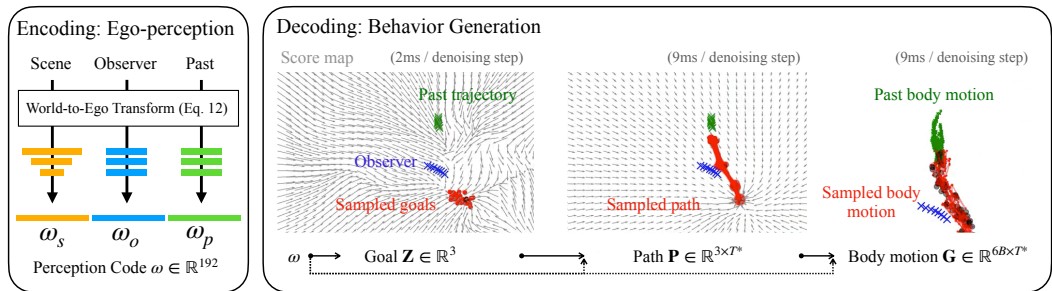

Figure 3: Pipeline for behavior generation. We first encode egocentric information into a perception code $\omega$ and then generate full body motion in a hierarchical fashion. We start by generating goals $\mathbf{Z}$ with low latency, and then generate a path $\mathbf{P}$ and body motion $\mathbf{G}$ conditioned on the previous node. Each node is represented by the gradient of its log distribution, trained with the denoising objectives (Eq. 9). Given $\mathbf{G}$, the dense deformation of an agent can be computed via blend skinning (Eq. 3).

Compared to concatenation, we observe better alignment between the path and the full body pose using the Control Unet.

.

**Ego-Perception Encoding.** To generate plausible interactive behaviors, we encode the world *egocentrically* perceived by the agent, and use it to condition the behavior generation. We use the reconstructed environment $\mathbf{T}$ and the observer $\boldsymbol{\xi}$ as a proxy of the world, and transform them to the egocentric coordinate of the agent,

$$\boldsymbol{\xi}^{s \to a} = \mathbf{G}_{b=0}^{-1}\boldsymbol{\xi}, \quad \mathbf{T}^{s \to a} = \mathbf{G}_{b=0}^{-1}\mathbf{T} \tag{12}$$

Transforming the world to the egocentric coordinates avoids over-fitting to specific locations of the scene (Tab. 2). To encode ego-perception of the scene, we querying feature values from $\psi_s$ with a 3D grid around the agent and extract a latent scene representation,

$$\omega_s = \mathrm{ResNet3D}_{\theta_\psi}(\psi_s). \tag{13}$$

where $\mathrm{ResNet3D}_{\theta_\phi}$ is a 3D ConvNet with residual connections, and $\omega_s \in \mathbb{R}^{64}$ represents the scene perceived by the agent. We encode the observer's motion in the past $T' = 0.8s$ seconds with

$$\omega_o = \mathrm{MLP}_{\theta_o}(\boldsymbol{\xi}^{s \to a}), \tag{14}$$

where $\omega_o \in \mathbb{R}^{64}$ represents the observer perceived by the agent. Accounting for the external factors from the "world" enables interactive behavior generation, where the motion of an agent follows the environment constraints and is influenced by the trajectory of the observer (Fig. 5).

**History Encoding.** We additionally encode the past motion of the agent in $T'$ seconds,

$$\omega_p = \mathrm{MLP}_{\theta_p}(\mathbf{G}_{b=0}^{s \to a}). \tag{15}$$

By conditioning on the past motion, we can generate long sequences by chaining individual ones.

## 4  Experiments

**Dataset.** We collect the a dataset that emphasizes the casual interactions of an agent with their familiar environment and the observer. It contains iPhone-captured RGBD video collections of 4 types of agents, including 26 videos of a cat, 3 videos of a dog, 2 videos of a bunny, and 2 videos of a human. The time span of the video capture ranges from 1 day to a month, and each video contains 30 seconds to 2 minutes of content. The dataset is curated to contain diverse motion of agents, including walking, lying down, eating, as well as diverse interaction patterns with the environment, including following the camera, sitting on a coach, etc. Please refer to the supplement for more details.

### 4.1  4D Reconstruction of Agent & Scene

**Implementation Details.** We extract frames from the videos at 10 FPS, and use off-the-shelf models to produce augmented image measurements, including object segmentation [68], optical flow [63],

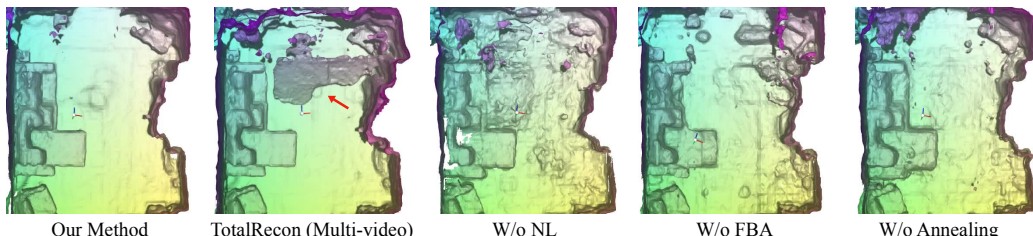

| Our Method | TotalRecon (Multi-video) | W/o NL | W/o FBA | W/o Annealing |

Figure 4: **Comparison on multi-video scene reconstruction**. We show a top-down visualization of the reconstructed scene using the bunny dataset. Compared to TotalRecon that does not register multiple videos, ATS produces higher-quality scene reconstruction. Neural localizer and featuremetric losses are shown important for camera registration. Scene annealing is important for reconstructing high-quality scenes from limited views in a video.

DINOv2 features [40]. We use AdamW to first optimize the environment with featuremetric loss for 30k iterations, and then jointly optimize the environment and agent for another 30k iterations with a combination of optical flow, silouette, and featuremetric losses. Optimization takes roughly 24 hours. 8 A100 GPUs used to optimize 26 videos (for the cat data), and 1 A100 GPU is used in a 2-3 video setup (for dog, bunny, and human data).

**Results.** We run 4D reconstruction on all video sequences and report the results qualitatively. A visual comparison on scene registration is shown in Fig. 2. Without the ability to register multiple videos, TotalRecon produces protruded and misaligned structures (as pointed by the red arrow). In contrast, our method reconstructs a single coherent scene. With featuremetric alignment (FBA) alone but without a good camera initialization from neural localization (NL), our method produces inaccurate reconstruction due to global misalignment in cameras poses. Removing FBA while keeping NL, the method fails to accurately localize the cameras and produces noisy scene structures. Finally, removing scene annealing procures lower quality scene structures due to lack of training views. A visual comparison with TotalRecon (Single Video) is shown in Fig. 8, where we show that multiple videos helps reconstructing a higher-quality agent, and a more complete scene.

## 4.2 Interactive Behavior Prediction

**Dataset.** We use the cat dataset for quantitative evaluation, where the data are split into a training set of 22 videos and a validation set of 4 videos. The validation set is representative of three dominant motion patterns of the agent: (1) trying to engage with the observer, (2) exploring the space and (3) performing activities while not paying attention to the observer.

**Implementation Details.** To train the behavior model, we slice the reconstructed trajectory in the training set into overlapping window of $6.4$s, resulting in 12k data samples. We use AdamW to optimize the parameters of the scores functions $\{\theta_{\mathbf{Z}}, \theta_{\mathbf{P}}, \theta_{\mathbf{G}}\}$ and the ego-perception encoders $\{\theta_\psi, \theta_o, \theta_p\}$ for 120k steps with batch size 1024. Training takes 10 hours on a single A100 GPU.

**Metrics.** The behavior of an agent can be evaluated along multiple axes, and we focus on goal, path, and body motion prediction. For goal prediction, we use a combination of displacement error (DE) and minimum displacement error (minDE) [7]. The evaluation asks the model to produce K=64 samples. DE computes the avarage distance of the samples to the ground-truth, and minDE finds the one closest to the ground-truth to compute the distance. For path and body motion prediction, we use average displacement error (ADE) and minimum average displacement error (minADE), which are similar to goal prediction, but additionally averages the distance over path and joint locations before taking the min. When evaluating path prediction and body motion prediction, the output is conditioned on the ground-truth goal and path respectively.

**Comparisons.** We re-purpose related methods and adapt them to our new setup of interactive behavior prediction of animal agents. The quantitative results are shown in Tab. 2. To predict the goal of an agent, classic methods build statistical models of how likely an agent visits a spatial location of the scene, referred to as location prior [26, 76]. Given the extracted 3D trajectories of an agent in the egocentric coordinate, we build a 3D preference map over 3D locations as a histogram, which can be turned into probabilities and used to sample goals. Since this method does not take into account

Table 2: **Evaluation of interactive behavior prediction.** We separately evaluate goal, path, and full body motion prediction. Metrics are displacement errors (DE) in meters and the lower the better. FaF [33] is re-purposed and re-trained with our data.

| Method | Goal: minDE | Goal: DE | Path: minADE | Path: ADE | Body: minADE | Body: ADE |
|---|---|---|---|---|---|---|
| Location prior [76] | 0.575 | 2.134 | N.A. | N.A. | N.A. | N.A. |
| FaF [33] | N.A. | 1.200 | N.A. | 0.057 | N.A. | 0.265 |
| ATS (Ours) | **0.395** | 1.299 | **0.006** | **0.007** | 0.226 | 0.234 |
| w/o observer $\omega_o$ | 0.525 | 1.586 | **0.006** | **0.007** | 0.225 | 0.234 |
| w/o scene $\omega_s$ | 0.702 | **1.058** | **0.006** | **0.007** | 0.225 | 0.234 |
| w/o egocentric | 0.639 | 1.424 | 0.025 | 0.034 | **0.212** | **0.222** |

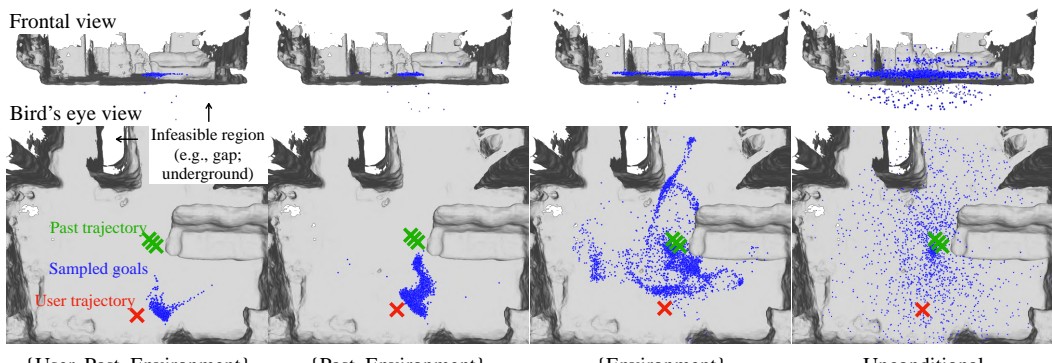

Figure 5: Analysis of conditioning signals. We show results of removing one conditioning signal at a time. Removing observer conditioning and past trajectory conditioning makes the sampled goals more spread out (e.g., regions both in front of the agent and behind the agent); removing the environment conditioning introduces infeasible goals that penetrate the ground and the walls.

of the scene and the observer, it fails to accurately predict the goal. We then re-purpose FaF [33] (Fast-and-Furious), a data-driven approach for motion forecasting to our task. FaF takes the same input as ATS but regresses the goal, path, and body poses. It produces worse results than ATS for all metrics since directly regressing the target treats the underlying distribution as a unit-variance Gaussian and fails to account for the multi-modal nature of agent behaviors.

**Analysing Interactions.** We analyse the agent's interactions with the environment and the observer by removing the conditioning signals and study their influence on behavior prediction. In Fig. 5, we show that by gradually removing conditional signals, the generated goal samples become more spread out. In Tab. 2, we drop one of the conditioning signals at a time. Dropping the observer conditioning increases the error in goal prediction, indicating observer's trajectory is helpful goal prediction. Dropping the environment conditioning produces worse results on goal prediction (minDE: 0.395 vs 0.702) as well. Surprisingly, it does not affect path prediction. We posit that the scenarios in the test set are too simple. Conditioned on ground-turth goals, it performs well even without environment conditioning. Finally learning behavior generation in the world coordinates performs worse for all metrics since it over-fits to specific locations in the scene.

## 5 Conclusion

We have presented a framework for learning interactive behavior of agents grounded in natural environments. To achieve this, we turn multiple casually-captured video recordings into complete 4D reconstructions including the agent, the environment, and the observer. Such data collected over a long time period allows us to learn a behavior model of the agent that is reactive to the observer and respects the environment constraints. We validate our design choices on casual video collections, and show better results than prior work for 4D reconstruction and interactive behavior prediction.

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

# A  Additional Implementation Details

**Model Architecture.** The score function of the goal is implemented as 6-layer MLP with hidden size 128. The the score functions of the paths and body motions are implemented as 1D UNets taken from MDM [57]. The sampling frequency is set to be $0.1$s, resulting a sequence length of $56$. The environment encoder is implemented as a 6-layer 3D ConvNet with kernel size 3 and channel dimension 128. The observer encoder and history encoder are implemented as a 3-layer MLP with hidden size 128.

We use a linear noise schedule at training time and $50$ denoising steps. At test time, each goal denoising step takes 2ms and each path/body denoising step takes 9ms on a GeForce RTX 3090 GPU.

**Data Collection.** We collect RGBD videos using an iPhone, similar to TotalRecon [52]. To train the neural localizer, we use Polycam to take the walkthrough video and extract a textured mesh. For behavior capture, we use Record3D App to record videos and extract color images and depth images.

# B  Additional Results

**Histogram of Agent / Observer Visitation.** We show final camera and agent registration to the canonical scene in Fig. 6. The registered 3D trajectories provides statistics of agent's and user's preference over the environment.

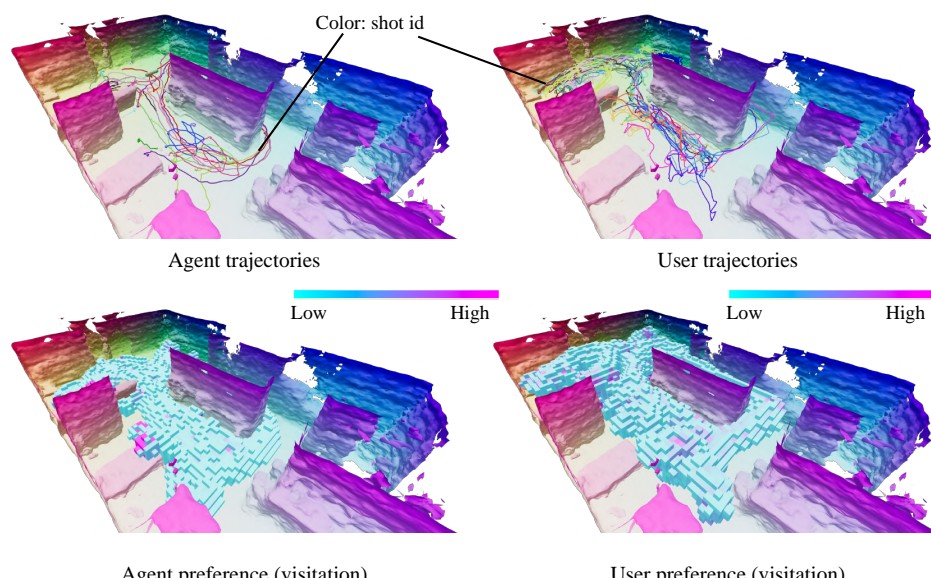

Figure 6: Given the 3D trajectories of the agent and the user accumulated over time (top), one could compute their preference represented by 3D heatmaps (bottom). Note the high agent preference over table and sofa.

**Varying Observer's Motion.** We find that various interactive behaviors can be generated by conditioning the model on different observer motion. The results are shown in Fig. 7.

**Comparison to TotalRecon.** In the main paper, we compare to TotalRecon on scene reconstruction by providing it multiple videos. Here, we include additional comparison in their the original single video setup. We find that TotalRecon fails to build a good agent model, or a complete scene model given limited observations, while our method can leverage multiple videos as inputs to build a better agent and scene model. The results are shown in Fig. 8.

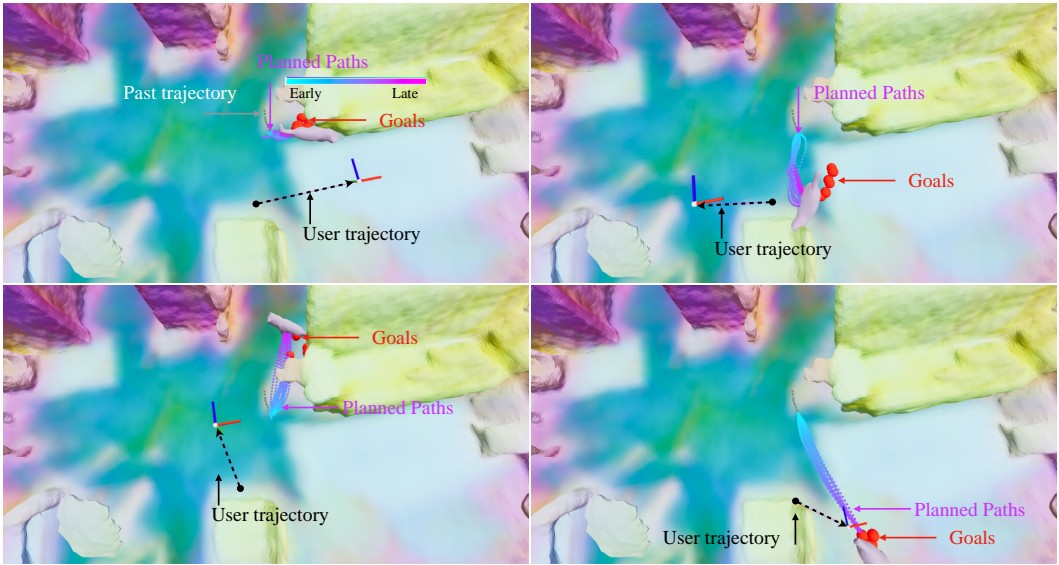

Figure 7: Interactive behavior simulation with user conditioning. By changing the trajectory of the user, one could influence the behavior of the agent. Given different control inputs, the agent may follow the user or run away from the user.

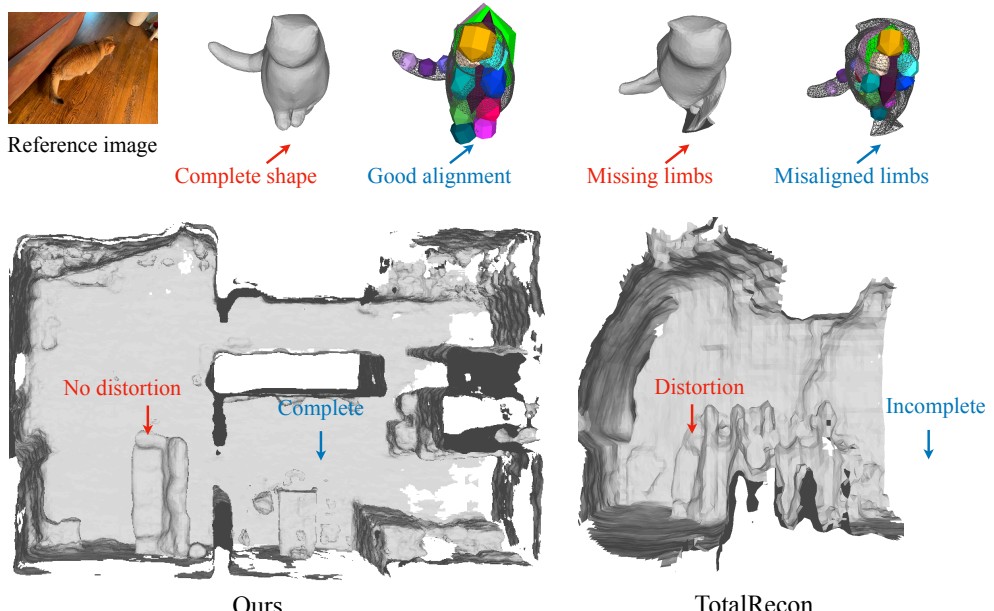

Figure 8: Qualitative comparison with TotalRecon [52] on 4D reconstruction. Top: reconstruction of the agent at at specific frame. Total-recon produces shapes with missing limbs and bone transformations that are misaligned with the shape, while our method produces complete shapes and good alignment. Bottom: reconstruction of the environment. TotalRecon produces distorted and incomplete geometry (due to lack of observations from a single video), while our method produces an accurate and complete environment reconstruction.

## C  Limitations and Future Works

**High-level Behavior.** The current ATS model is trained with time-horizon of $T^* = 6.4$ seconds. We observe that the model only learns mid-level behaviors of an agent (e.g., trying to move to a destination; staying at a location; walking around). We hope incorporating a memory module and training with longer time horizon will enable learning higher-level behaviors of an agent.

**Scaling-up.** As indicated by the experimental results, the goals sampled from ATS may fail to cover the actual goal when evaluated on the (unseen) test data. This raises safety concerns when using ATS for the prediction task (e.g., predicting the behavior of pedestrains in autonomous driving). One potential solution of improving the generalization ability is to collect more diverse behavior data from in the wild videos, or leverage "large" video priors trained on internet-scale videos.

**Multiple Agents.** We show results of learning behavior models of a single agent, but our method for 4D reconstruction and interactive goal-driven behavior modeling is not limited to a single agent. We leave learning multi-agent behavior simulation from videos as future work.

**Physical Interactions.** Our method reconstructs and generates the kinematics of an agent, which may produce physically-implausible results (e.g., penetration with the ground and foot sliding). One promising way to deal with this problem is to add physics constraints to the reconstruction and motion generation [67, 71].

**Environment Reconstruction.** To build a complete reconstruction of the environment, we register multiple videos to a shared canonical space. However, the transient structures (e.g., cushion that can be moved over time) may not be reconstructed well due to lack of observations. One potential solution of reconstructing these transient structures is to combine generative image priors with the reconstruction pipeline [60].

## D  Social Impact

Our method is able to learn interactive behavior from videos, which could help build simulators for autonomous driving, gaming, and movie applications. It is also capable of building personalized behavior models from casually collected video data, which can benefit users who do not have access to a motion capture studio. On the negative side, the behavior generation model could be used as "deepfake" and poses threats to user's privacy and social security.

