# OpenReview forum: "Agent-to-Sim: Learning Interactive Behavior from Casual Videos"
_NeurIPS.cc/2024/Conference — Submitted to NeurIPS 2024_

### Official Review · Reviewer_xDUi · 2024-07-10

**Soundness:** 3
**Presentation:** 4
**Contribution:** 3
**Rating:** 7
**Confidence:** 3

**Summary:**

The paper presents ATS (Agent-To-Sim), a framework to enable agent behavior modeling from multiple casual video captures in indoor scenarios captured during long spans of time. The proposed pipeline consists in (1) 4D reconstruction of the scene geometry and observer and agent motion, and (2) controllable agent behavior learning and generation.

For the first stage, multi-video registration uses coarse-to-fine registration to globally align the cameras to a shared canonical space derived from DINOv2 per-frame features (initialized with a walkthrough clip of the environment) and then jointly optimizes the 3D structures while adjusting the cameras locally with novel featuremetric losses (which makes the optimization robust to changes of lighting and appearance and improves alignment accuracy) and standard photometric and regularization losses. With the proposed (annealed) swapping of latent per-video codes during optimization, missing information is shared across videos, while video-specific details are kept.

For the controllable agent behavior modeling, in order to generate plausible interactive behaviors, the generated behavior conditions on an encoding of the scene, observer, and past from the agent's egocentric perspective, which avoids overfitting to specific locations in the scene. Then, the ego-perception-conditioned generation of full body motion proceeds hierarchically via diffusion: Generated goals Z condition generated paths P, which finally condition generated body motions G.

The included experiments reflect the quality of the 4D reconstructions achieved by the proposal, the improvements in displacement errors compared to two baselines (as well as ablations of the proposed method), and a qualitative analysis of the effects of the behavior conditioning signals.

**Strengths:**

- Great technical achievement to reconstruct agent behavior in indoor settings, exploiting the shared information across different videos captured at different times via robust alignment based on semantic features from foundational image models (DINOv2) and diffusion-based short-term hierarchical motion generation.
- Plausible long-horizon generation of agent motion for different bodies, conditioned on the environment, observer, and past trajectory.
- Despite the complexity of the system, the description is relatively brief and complete, whig, along with the rest of the paper, is excellently written.

**Weaknesses:**

- The paper focuses on environment-aware motion of agents in the presence of a (human) observer. Even if out of scope for this paper, it would be interesting to discuss more complex agent-environment interactions (see my questions below).
- I believe the current experiments use a small number of environments/scenes, which makes it hard to justify considering the system for larger-scale deployment, but I'll be happy to update my score if the authors correct me.

**Questions:**

- In the appendices (L.507) and implementation details (L. 245) the training time horizon is 6.4 seconds, but in L.181 the motion modeling sets a horizon of 5.6 seconds. Is my understanding correct that, for each window, the first 0.8 seconds are used as previous trajectory and the remainder 5.6 seconds as targets?
- How is an additional (moving) agent in the scene (e.g. a person or another animal moving in the background) currently handled by the 4D agent-observer-scene modeling method described for a single agent?
- Are there many examples of videos not reconstructible due to notable changes in the scene layout (e.g. movement of large furniture) in the captured data? If so, how is it handled? I did not see any reference to this issue in the original manuscript, but I think it is a reasonable discussion when attempting to employ this framework at scale.
- How could this framework be extended towards modeling complex interactions with the environment (e.g. opening a door, sitting on a chair after moving it, etc.)?
- How many scenes and different agents are used in training and validation? Do they specific agents overlap across splits, or just the types of agents?

**Limitations:**

The authors reasonable address limitations and social impact in the appendices.

---

> ### Author Rebuttal · Authors · 2024-08-06
>
> Thanks for your constructive feedback! Below please find our responses to your questions and comments.
>
> **Q1 The paper focuses on environment-aware motion of agents in the presence of a (human) observer. Even if out of scope for this paper, it would be interesting to discuss more complex agent-environment interactions. How is an additional (moving) agent in the scene (e.g. a person or another animal moving in the background) currently handled by the 4D agent-observer-scene modeling method described for a single agent?**
>
> Our model is designed to handle interactions of a single agent with the observer and the scene. The other agents that appear in the videos are treated as part of the scene. However, due to the static scene assumption (Eq. 1), those moving agents are either averaged out or not well reconstructed (please see Fig. D in the [rebuttal pdf](https://openreview.net/attachment?id=abfRu0bgF7&name=pdf) for a visual example). Solving re-identification and multi-object tracking [E] in 3D will enable introducing multiple agents, which is an exciting future work.
>
> [E] Rajasegaran, Jathushan, et al. "Tracking people by predicting 3d appearance, location and pose." CVPR 2022.
>
> **Q2 Are there many examples of videos not reconstructible due to notable changes in the scene layout (e.g. movement of large furniture) in the captured data? If so, how is it handled? I did not see any reference to this issue in the original manuscript, but I think it is a reasonable discussion when attempting to employ this framework at scale.**
>
> Thanks for the suggestion. As shown in Fig. D of the [rebuttal pdf](https://openreview.net/attachment?id=abfRu0bgF7&name=pdf), our method fails to reconstruct notable layout changes when they are only observed in a few views, e.g., the cushion and the large boxes (left) and the box (right). Leveraging generative image prior to in-paint the missing regions is a promising direction to tackle this problem [F].
>
> [F] Weber, Ethan, et al. "Nerfiller: Completing scenes via generative 3d inpainting." CVPR. 2024.
>
> **Q3 How could this framework be extended towards modeling complex interactions with the environment (e.g. opening a door, sitting on a chair after moving it, etc.)?**
>
> To reconstruct complex interactions with the environment, one idea is to extend the scene representation to be hierarchical (represented as a kinematic tree), such that it consists of articulated models of interactable objects [G, H, I]. To generate plausible interactions between the agent and the scene (e.g., opening a door), we can extend the agent state $G$ to include both the agent and the articulated objects (e.g., door), and learn a behavior generator to generate their trajectory jointly [J].
>
> [G] Song, Chaoyue, et al. "REACTO: Reconstructing Articulated Objects from a Single Video." CVPR. 2024.
>
> [H] Liu, Jiayi, Ali Mahdavi-Amiri, and Manolis Savva. "Paris: Part-level reconstruction and motion analysis for articulated objects." ICCV. 2023.
>
> [I] Wei, Fangyin, et al. "Self-supervised neural articulated shape and appearance models." CVPR. 2022.
>
> [J] Li, Jiaman, et al. "Controllable human-object interaction synthesis." ECCV 2024.
>
> **Q4 How many scenes and different agents are used in training and validation? Do specific agents overlap across splits, or just the types of agents?**
>
> We demonstrate our approach on four types of agents with different morphology, i.e., cat, human, dog, and bunny, in three different scenes, where human and cat share the same scene. We train an instance-specific model for each agent, and the data is not mixed up across agents. For quantitative evaluation of behavior generation, we report the performance of the cat agent on the held-out test sequences of the same cat. Training a model across different agent identities and types will be an interesting future work.
>
> **Q5 The current experiments use a small number of environments/scenes, which makes it hard to justify considering the system for larger-scale deployment.**
>
> For the cat dataset, we use 26 video clips over the span of a month, which is not super large-scale but we believe this is an important step to go beyond a single video. The major difficulty towards large-scale deployment is the efficiency and robustness of 4D reconstruction algorithms. In terms of robustness, we showed a meaningful step towards scaling up 4D reconstruction by neural initialization (Eq. 6). We believe this paper is a step toward large scale deployment, and we will release code for reproducibility and make it easy to follow up.
>
> **Q5 In the appendices (L.507) and implementation details (L. 245) the training time horizon is 6.4 seconds, but in L.181 the motion modeling sets a horizon of 5.6 seconds. Is my understanding correct that, for each window, the first 0.8 seconds are used as previous trajectory and the remainder 5.6 seconds as targets?**
>
> You are totally correct about this!

---

> > ### Comment · Reviewer_xDUi · 2024-08-12
> >
> > Thank you for the responses and clarifications. I think the paper is nice to read and shows promising results, so I am keeping my score.

---

### Official Review · Reviewer_C9Gr · 2024-07-13

**Soundness:** 3
**Presentation:** 2
**Contribution:** 3
**Rating:** 6
**Confidence:** 2

**Summary:**

This paper discusses using an iPhone's RGBD camera to collect several hours of videos within a room over a time span of one month. Through these multi-view videos, a 4D reconstruction of the room is generated. A collection of rigid bodies is used to simulate agents (such as cats, dogs, etc.) in the room. Utilizing goal-conditional path generation technology, users can ultimately control the movement of these agents by setting goals.

**Strengths:**

1. The video presented in this paper is very effective; it reconstructs 4D video from a single view and reconstructs a complete room from multiple views.
2. In addition to reconstruction, the paper also discusses how to control the movement of the agent through goal-condition path generation.
3. Intuitively, I think this is a good paper and may inspire researchers in the field of 4D reconstruction.

**Weaknesses:**

1. While I am not an expert in 4D reconstruction, I find the presentation of this paper rather unclear, particularly the methodology section, which is extremely difficult to understand. My confusion began around lines 126-127. What are the color and feature descriptors of the video? I later noticed that ψ is described as the DINOv2 [40] feature of the input image. So, is ψ a feature of an image? How to obtain it? The paper should clarify this. Additionally, what is X, and is it a point cloud obtained from a mobile phone? If so, how does the point cloud acquire its color in Equation 2?

2. I suggest using a table to explain each symbol in detail. If the explanation of a symbol requires context from the paper, ensure it is as understandable as possible. For technical terms, provide detailed explanations within the paper. A comprehensive symbol table in the appendix would significantly enhance the paper's clarity.

3. The paper lacks detailed quantitative experiments to demonstrate the effectiveness of the method.

**Questions:**

What is the practical use of this work?

**Limitations:**

The authors adequately addressed the limitations and potential negative societal impact.

---

> ### Author Rebuttal · Authors · 2024-08-06
>
> Thanks for your constructive feedback! We added a table of notations to improve the clarity, and we will expand the explanation of individual symbols in the paper.
>
> | Notation      | Description                                                                                          |
> |-------------|------------------------------------------------------------------------------------------------------|
> | **Global Symbols** |
> | $B$         | The number of bones of an agent. By default $B = 25$.                                                |
> | $M$         | The number of videos.                                                                                |
> | $N_i$       | The number of image frames extracted from video $i$.                                                 |
> | $I_i$       | The sequence of color images {$\{I_1, \ldots, I_{N_i}\}$} extracted from video $i$.                    |
> | $\psi_i$    | The sequence of DINOV2 feature images {$\{\psi_1, \ldots, \psi_{N_i}\}$} extracted from video $i$.     |
> | $T_i$       | The length of video $i$.                                                                             |
> | $T^*$       | The time horizon of behavior diffusion. By default $T^* = 5.6s$.                                     |
> | $T'$        | The time horizon of past conditioning. By default $T' = 0.8s$.                                       |
> | $Z \in \mathbb{R}^3$     | Goal of the agent, defined as the location at the end of $T^*$.                         |
> | $P \in \mathbb{R}^{3 \times T^*}$ | Path of the agent, defined as the root body trajectory over $T^*$.              |
> | $G \in \mathbb{R}^{6B \times T^*}$ | Pose of the agent, defined as the 6DoF rigid motion of bones over $T^*$.      |
> | $\omega_s \in \mathbb{R}^{64}$ | Scene code, representing the scene perceived by the agent.                        |
> | $\omega_o \in \mathbb{R}^{64}$ | Observer code, representing the observer perceived by the agent.                  |
> | $\omega_p \in \mathbb{R}^{64}$ | Past code, representing the history of events happened to the agent.              |
> | **Learnable Parameters:** | **4D Reconstruction**|
> $T$         | Canonical NeRFs, including a scene MLP and an agent MLP.                                   |
> | $\beta_i \in \mathbb{R}^{128}$ | Per-video code that allows NeRFs to represent variations across videos.            |
> | $\mathcal{D}$ | Time-varying parameters, including {$\{\xi, G, W\}$}.                                                |
> | $\xi_t \in SE(3)$ | The camera pose that transforms the scene to the camera coordinates at $t$.                     |
> | $G^b_t \in SE(3)$ | The transformation that moves bone $b$ from its rest state to time $t$ state.                |
> | $W \in \mathbb{R}^B$ | Skinning weights of a point, defined as the probability of belonging to bones.               |
> | $f_\theta$ | PoseNet that takes a DINOV2 feature image as input and produces camera pose.                         |
> | **Learnable Parameters:** | **Behavior Generation**|
> | MLP$_{\theta_z}$     | Goal MLP that represents the score function of goal distributions.                                   |
> | ControlUNet$_{\theta_p}$| Path UNet that represents the score function of path distributions.                               |
> | ControlUNet$_{\theta_G}$| Pose UNet that represents the score function of pose distributions.                               |
> | ResNet3D | Scene perception network that produces $\omega_s$ from 3D feature grids.                                 |
> | MLP$_{\theta_o}$     | Observer MLP that produces $\omega_o$ from observer’s past trajectory in $T'$.                                |
> | MLP$_{\theta_p}$     | Past MLP that produces $\omega_p$ from agent’s past trajectory in $T'$.                                       |
>
> Below please find our responses to your questions and comments.
>
> **Q1 What are the color and feature descriptors of the video?**
>
> Given a video $i$ with $N_i$ frames, we extract color images {$\{I_1, \ldots, I_{N_i}\}$} and compute their dense features descriptors {$\{\psi_1, \ldots, \psi_{N_i}\}$} using a pre-trained DINOv2 network.
>
> **Q2 What is X, and is it a point cloud obtained from a mobile phone? If so, how does the point cloud acquire its color in Equation 2?**
>
> ${\bf X}$ in Eq. 1-3 are not point clouds. They are continuous 3D coordinates used to the define density and color of NeRFs in its implicit representation. The color at a location ${\bf X}$ can be queried by evaluating the MLP in Eq. 2. The parameters of the MLP are learned via differentiable rendering optimization in Eq. 7. We will clarify in the paper.
>
> **Q3 The paper lacks detailed quantitative experiments to demonstrate the effectiveness of the method.**
>
> We provided additional quantitative comparisons and analysis in the [global response](https://openreview.net/forum?id=fzdFPqkAHD&noteId=abfRu0bgF7) Table A-C, specially on camera localization, 4D reconstruction, and behavior learning. We found that our design choices are necessary to achieve good performance.
>
> **Q4 What is the practical use of this work?**
>
> Our goal is to learn “world models” that one can interact with from videos. This is a fundamental question that has practical application in generating contents for VR/AR, as well as robot learning with plausible agent simulation. For VR/AR applications, our approach enables generating data-driven agents that can interact with humans and scenes in a realistic manner. For robotics, the learned realistic behavior simulation can be used to pretrain robot policies that have a smaller sim-to-real gap, before adapting to the real world.

---

> > ### Comment · Reviewer_C9Gr · 2024-08-09
> >
> > Thank you for your response. The table is very clear and addresses most of my concerns. I would suggest that, in future versions, this information be included in the appendix, where it can be expanded in greater detail, given that space is not as limited there. This addition would greatly enhance the reader's understanding of the paper.
> >
> > While it is challenging to reproduce the results based solely on the paper's description, the availability of the code is a significant asset. If the code is complete, I believe that with improvements in the clarity of the writing and the inclusion of more detailed explanations in the appendix, this paper could be strong enough for publication in NeurIPS. However, my primary concern is that the current presentation makes the paper somewhat difficult to follow, which could impact its accessibility to a broader audience.
> >
> > My remaining question is whether the authors could restate the paper’s task, objectives, inputs and outputs, datasets, key modules, and the corresponding inputs and outputs for those key modules in a way that is more accessible to readers. I believe these elements could be added to the appendix to further aid reader comprehension.

---

> > > ### Author Response · Authors · 2024-08-09
> > >
> > > Thanks for your response and additional feedback. We provide the requested elements below and will improve the presentation accordingly. Please kindly let us know if anything is missing or unclear.
> > >
> > > **Task and Objectives.** We develop a method to learn interactive behavior models of agents from casual videos captured over a long time horizon. The objectives include:
> > > - **Casual 4D reconstruction**: Enabling low-cost capture of agent’s shape, motion, and interactive behavior signals from casual videos (e.g., captured with an iPhone);
> > > - **Interactive behavior modeling**: Learning behavior of agents interacting with the environment and the observer; and
> > > - **Flexible representation**: Extending behavior learning to broader agent categories, such as animals,
> > >
> > > which will ultimately contribute to VR/AR and Robotics via generating interactive contents for VR/AR, as well as robot learning with plausible agent simulation.
> > >
> > > **Dataset.** We demonstrate our approach on four types of agents with different morphology, i.e., cat, human, dog, and bunny, in three different scenes, where human and cat share the same scene. Here is a breakdown of the data we used
> > > |        | # Videos | # Frames | # Days | Time span (days) |
> > > |--------|--------------|--------------|------------|------------------|
> > > | Cat    | 26           | 15391        | 9          | 37               |
> > > | Human  | 5            | 5668         | 2          | 4                |
> > > | Dog    | 3            | 4330         | 1          | 1                |
> > > | Bunny  | 2            | 1080         | 1          | 1                |
> > >
> > >
> > > **Input/output**. We provide the input/output of the global system and key submodules below. We integrated the above into a pipeline figure and will add it to the appendix.
> > >  - **Global Input/output**
> > >    - Input: A walk-through video of the environment and a video collection of target agent.
> > >    - Output: An interactive behavior generator of the agent.
> > >
> > > - **Neural localization** (Sec 3.2, L154-165)
> > >   - Input: Neural localizer  $f_\theta$ and the video collection of the agent.
> > >   - Output: Camera poses for each video frame.
> > >
> > > - **4D Reconstruction with feature-metric alignment** (Sec 3.2, L167-176)
> > >   - Input: Video collection of the agent and corresponding camera poses.
> > >   - Output: Reconstruction of the geometry ${\bf T}$, agent motion ${\bf G}$ and observer motion ${\boldsymbol \xi}$.
> > >
> > > - **Behavior learning** (Sec. 3.3)
> > >   - Input: Reconstruction of the scene geometry ${\bf T}$, agent motion ${\bf G}$ and observer motion ${\boldsymbol \xi}$.
> > >   - Output: An interactive behavior generator of the agent.
> > >
> > > - **Behavior generation** (Fig. 3)
> > >   - Input: Ego-centric scene feature grid, agent's past trajectory over horizon $T'=0.8s$, observer's past trajectory over $T'=0.8$.
> > >   - Output: Goal, path, and a sequence of full body motion of the agent over $T^*=5.6s$.

---

> > > > ### Comment · Reviewer_C9Gr · 2024-08-11
> > > >
> > > > Thank you for your response. My understanding of the paper is that it first involves learning to predict the camera pose in the video, followed by 4D scene reconstruction. Then, the reconstructed information is used to learn the agent's movement. During the inference phase (behavior generation), the agent's behavior is simulated by specifying a goal. I admit that this approach is novel to me. The advantage of 4D reconstruction lies in having an existing 3D model, which might reduce the hallucination problem that can occur in purely video-based generation. It would be great if the authors could make the method clearer in future versions. Considering that my questions have been addressed, I have increased my rating.
> > > >
> > > > My remaining concern is that, despite the presence of an intrinsic 3D scene and objects in this method, how can we ensure its scalability to new scenes and objects? How can it be realistically applied to VR/AR or robot learning? From the perspective of robot learning, this method seems quite far from real-world applications. I would be happy to discuss with the authors how this method could be realistically applied to policy learning. Particularly in robotics, where there are many manipulable objects, it might not be feasible to use a simple setup of 25 spheres for simulation, as this method may not be scalable to new manipulable objects. However, overall, while this approach might not be very practical at the moment, it does have some inspirational value.

---

> > > > > ### Author Response · Authors · 2024-08-12
> > > > >
> > > > > Thank you for sharing additional insights! We are happy to discuss how ATS can be applied to robot learning.
> > > > >
> > > > > **Locomotion Policy Learning.**
> > > > >
> > > > > For learning locomotion policy for humanoid and quadruped robots, ATS can already be used without architectural changes, in terms of
> > > > > - Offline Generation: The behavior generator can be used to produce training data as the input to existing motion imitation algorithms, for instance [A, Sec. V];
> > > > > - Online Generation: During deployment time, the behavior model can be used d to generate plausible trajectories for position-based control or MPC (model predictive control), similar to diffusion policy [B, Sec IV. B].
> > > > >
> > > > > **Manipulation Policy Learning.**
> > > > >
> > > > > We need to make a few changes to extend ATS to manipulation tasks involving complex object interactions,
> > > > > - Scene representation: During differentiable rendering, we could leverage object-compositional scene representation [C], such that the scene consists of interactable objects.
> > > > > - Object-interaction generation: To generate plausible interactions between the agent and the objects (e.g., picking up a tool), we can extend the agent state to include both the agent and the objects, and learn a behavior model to generate their trajectory jointly, similar to [D].
> > > > >
> > > > > [A] Peng, Xue Bin, et al. "Learning agile robotic locomotion skills by imitating animals." RSS 2020.
> > > > >
> > > > > [B] Chi, Cheng, et al. "Diffusion policy: Visuomotor policy learning via action diffusion." RSS 2023.
> > > > >
> > > > > [C] Wu, Qianyi, et al. "Objectsdf++: Improved object-compositional neural implicit surfaces." CVPR. 2023.
> > > > >
> > > > > [D] Li, Jiaman, et al. "Controllable human-object interaction synthesis." ECCV 2024.

---

> > > > > > ### Comment · Reviewer_C9Gr · 2024-08-13
> > > > > >
> > > > > > But the trajectory you mentioned here is not executable actions, but rather the 3D pose trajectory of an agent. I believe there is a significant gap. However, thank you for the references provided; I look forward to seeing real applications of this work, as genuine challenges will aid in adjusting your research questions. Considering this, I will maintain a 'weakly accept' as my final score.

---

### Official Review · Reviewer_Kj9h · 2024-07-13

**Soundness:** 3
**Presentation:** 3
**Contribution:** 3
**Rating:** 6
**Confidence:** 1

**Summary:**

This paper presents Agent-to-Sim, an approach to learn a 3D agent in a 3D environment from casual videos of the same agent captured over a long horizon. ATS first conducts 4D spatio-temporal reconstruction from the set of videos, including a deformable agent, the background scene, and a moving observer. This is done with a coarse-to-fine video registration method. Then, given the 4D reconstruction, ATS learns a hierarchical diffusion model over the agent's goal, path, and pose trajectories.  The overall approach is tested on a dataset of iPhone videos for over several types of agents and motion patterns.

**Strengths:**

- I am not a subject matter expert in this field. However, the paper was clear and well-written such that even a non-expert like myself can understand the proposed high-level approach. The attached supplementary materials give a great visual overview of the paper.
- The paper outlines several limitations of the proposed approach and future directions to address them. The limitations are meaningful and help the reader better understand the problem setting, modelling assumptions, and future directions.
- The paper tackles a challenging problem on the path towards building scalable and realistic simulators.

**Weaknesses:**

- Certain technical details are not clear for readers unfamiliar with the related literature. This limits understanding and reproducibility. See questions.
- Evaluation of the method seems limited and is mostly limited to qualitative comparisons. I suppose this is inevitable given that ATS tackles a new problem setting than related work. However, it does limit the reader's ability to evaluate the significance of this methodology.
- For behavior generation evaluation, I don't understand why certain baselines were selected. In particular, FaF seems like a detection + multi-agent motion forecasting paper for self-driving, so it's not immediately clear how it can be adapted to this setting.

**Questions:**

1. What is the video code $ \beta $ and how is it used?
2. How are the ego-perception codes used in behavior generation?
3. What is B in L182?
4. Why is each module in behavior generation evaluated separately, conditioned on GT inputs? Since the task is behavior prediction, another natural evaluation setting seems to be an end-to-end evaluation setting comparing body motion prediction from ego-perception inputs. This would open up other ablation studies to understand the efficacy of the hierarchical model; e.g., by comparing against a non-hierarchical diffusion model.
5. In L191-196, how are the two diffusion models used? Are they combined to use a form of classifier-free guidance?

**Limitations:**

The authors have adequately addressed limitations and potential social impact of their work.

---

> ### Author Rebuttal · Authors · 2024-08-06
>
> Thanks for your constructive feedback! We plan to expand on details in the additional page of the final version as well as the appendix. In [the response to Reviewer C9Gr](https://openreview.net/forum?id=fzdFPqkAHD&noteId=cNiL3khmUC), we also added a table of notations to improve the clarity. Our code and data will be released for reproducibility and our goal is to allow researchers to continue working along this path.
>
> Below please find our responses to your questions and comments.
>
> **Q1 Evaluation of the method seems limited and is mostly limited to qualitative comparisons.**
>
> We provided additional quantitative comparisons and analysis in the [global response](https://openreview.net/forum?id=fzdFPqkAHD&noteId=abfRu0bgF7) Table A-C, specially on camera localization, 4D reconstruction, and behavior learning. We found that our design choices are necessary to achieve good performance.
>
> **Q2 Why is each module in behavior generation evaluated separately, conditioned on GT inputs? Since the task is behavior prediction, another natural evaluation setting seems to be an end-to-end evaluation setting comparing body motion prediction from ego-perception inputs. This would open up other ablation studies to understand the efficacy of the hierarchical model; e.g., by comparing against a non-hierarchical diffusion model.**
>
> Thanks for the great suggestion! We re-did the evaluation of the behavior prediction using the suggested end-to-end setup without using GT goals/paths. As a result, hierarchical out-performs one-stage by a large margin for all metrics. We posit hierarchical model makes it easier to learn individual modules. *Please see the [global response](https://openreview.net/forum?id=fzdFPqkAHD&noteId=abfRu0bgF7) for details.*
>
>
> **Q3 What is the video code 𝛽 and how is it used?**
>
> The video code 𝛽 is a 128-dimensional latent code that is concatenated to the fourier code as the input to NeRFs, similar to GIRAFFE [B]. We use this idea to represent scenes with slightly different layouts given a shared NeRF backbone.
>
> [B] Niemeyer, Michael, and Andreas Geiger. "Giraffe: Representing scenes as compositional generative neural feature fields." CVPR. 2021.
>
> **Q4 How are the ego-perception codes used in behavior generation?**
>
> The ego-perception codes are used as conditioning signals for behavior generation. Specifically, we concatenate the perception codes with the positional encoding of diffusion timesteps (representing the noise level $\sigma$) to predict the amount of noise. The predicted noise is then subtracted from the input noisy signal. This process is repeated for 50 times until a clean signal is obtained.
>
> **Q5 What is B in L182?**
>
> $B$ in $G \in \mathbb{R}^{6B \times T^*}$ is the number of bones of the agent. Each bone has 6 degrees-of-freedom including a center location (3DoF) and orientation (3DoF).
>
> **Q6 In L191-196, how are the two diffusion models used? Are they combined to use a form of classifier-free guidance?**
>
> Our behavior model consistents of 3 diffusion models, for goal, path, and full body motion generation respectively. Each diffusion model is trained with random dropout of the conditioning [C]. At test time, we classifier-free guidance to mix the conditional and unconditional score estimates with guidance-scale s = 2.5 following MDM [57].
>
> To enable precise user control for the path and full body models, we follow controlnet [72] and omnicontrol [62] that use two networks with identical architectures and dense skip connections in between. The first network receives the perception codes $\omega$ only; the second network receives additional control inputs (i,e., goal and path) and modulates the intermediate features of the first network. We found the ControlUNet architecture allows precise control when goal and path is provided by the user, as shown in Tab. D in the response to [Reviewer m6Ge](https://openreview.net/forum?id=fzdFPqkAHD&noteId=IUiGgGniSH).
>
> [C] Ho, Jonathan, and Tim Salimans. "Classifier-free diffusion guidance." arXiv preprint arXiv:2207.12598 (2022).
>
> **Q7 FaF seems like a detection + multi-agent motion forecasting paper for self-driving, so it's not immediately clear how it can be adapted to this setting.**
>
> This baseline represents goal, path, and full body motion as Gaussians, and learns to predict both the mean and variance of Gaussian distributions by minimizing the negative log-likelihood [D]. We implemented and trained it using the same data as ATS.
>
> This baseline was named as FaF because its input/output are close to FaF’s motion forecasting module. To avoid confusion, in the new [Table A](https://openreview.net/forum?id=fzdFPqkAHD&noteId=abfRu0bgF7) and [Table D](https://openreview.net/forum?id=fzdFPqkAHD&noteId=IUiGgGniSH), we renamed it as Gaussians.
>
> [D] Kendall, Alex, and Yarin Gal. "What uncertainties do we need in bayesian deep learning for computer vision?." NeurIPS 2017.

---

> > ### Comment · Reviewer_Kj9h · 2024-08-12
> >
> > Thank you for your response. I agree with Review m6Ge that the paper could benefit from more detailed exposition and evaluation. The authors have also agreed to improve the paper's exposition with an additional page and provided additional evaluation of its methodology in camera localization, 4D reconstruction, and behavior simulation. Considering this, I would like to maintain my rating and recommend accepting this paper.

---

### Official Review · Reviewer_m6Ge · 2024-07-18

**Soundness:** 2
**Presentation:** 2
**Contribution:** 3
**Rating:** 4
**Confidence:** 4

**Summary:**

The paper presents a method for learning interactive behaviors of various agents, including humans, cats, dogs and a bunny, by leveraging unstructured videos captured casually. The various videos are registered together in a common frame, offering a 4D reconstruction of the agent and the environment. Based on this reconstruction, the multi-modal distribution describing different agent behaviors is learned by using diffusion models and Control UNets.

**Strengths:**

The paper addresses the very challenging problems of learning agent behaviors from a collection of unstructured videos captured over different sessions. To learn interactive behaviors, both the trajectories of the agent and the surrounding environment need to be reconstructed, as to have relevant context of the behavior. Additionally, the motion of the camera/observer need to be reconstructed as well, to allow the registration of the videos in a common frame. As the videos are collected over a potentially large period of time, change in the environment can occur, complicating the tasks of registration and reconstruction.

The idea of using ego-perception encoding for the learning and generation of plausible interactive behaviors is another strong point. After the agent and the environment are reconstructed, ego-perception encoding is learning perception codes of the scene, the observer and past trajectory, factors that condition the generation of the agent's body motion.

Behavior generation considers the generation of the goal and the conditioned generation of the path, taking into account the goal.

**Weaknesses:**

There are numerous models employed in the proposed framework. Due to the limited space available, few details are provided about their motivation and their implementation. This makes both understanding of the work and its reproducibility very challenging.

A particular aspect which is not addressed in detail is the modeling of the agents, especially of animals like cats that are quite challenging due to their non-rigid nature. In particular, it is not clear how eq.2 is combined with eq.3, and why the same number of "bones" (b=25, L.137) is used for all agents. Also, the nature of G^b is not discussed in detail.

Additionally, details on how NeRF-type reconstructions are combined with feature descriptors, and how this helps in handling layout changes is not discussed in detail.

More examples like the previous can be given for different aspects covered in the paper, like camera localization (eq.6), scene alignment (eq.7) and behavior learning (eq.10 and 11). Each of these aspects would certainly require more space for describing in detail the corresponding models and support the relative claims in the experimental evaluation.

Regarding experimental evaluation in particular, only high-level results regarding the agent behavior prediction are provided, while it would be crucial to quantitatively assess the quality of 4D reconstruction and, importantly, to include a detailed ablative study.

Overall, although some very interesting ideas are proposed in this work, both for 4D reconstruction of agent behaviors and behavior learning and generation, I think that the paper is too densely packed without having enough space to describe the paper contributions in sufficient detail. In my view, even describing in detail one of the 4D reconstruction or agent behavior modeling parts alone would be challenging in the space available. This affects also the experimental evaluation, as not all claims are supported by the results.

### Minor comments
- L.35: "Such systems do not scale well"
- Figure 1, caption: incomplete sentence "conditioned different observer trajectories"
- L.88: "whiling accounts"
- L.113: what "longitudinal videos" are?
- Figure 3, caption: what does "low latency" means in this context?
- L.215: "we collect the a"

**Questions:**

- How are the agents, and especially the animals, modeled?
- What happens if a behavior is observed only once in the dataset or conversely, how many times need a behavior be observed to be included in the model?
- How robust is the method with respect to changes in the environment?

**Limitations:**

Limitations of the work are discussed in the Appendix.

---

> ### Author Rebuttal · Authors · 2024-08-06
>
> Thanks for your constructive feedback! Due to the complex nature of the problem, it is difficult to unpack all the details in the limited space. We plan to expand on details in the additional page of the final version as well as the appendix. In [the response to Reviewer C9Gr](https://openreview.net/forum?id=fzdFPqkAHD&noteId=cNiL3khmUC), we also added a table of notations to improve the clarity. Our code and data will be released for reproducibility and our goal is to allow researchers to continue working along this path.
>
> Below please find our responses to your questions and comments.
>
> **Q1 How are the agents, and especially the animals, modeled? A particular aspect which is not addressed in detail is the modeling of the agents, especially of animals like cats that are quite challenging due to their non-rigid nature.**
>
> We use the bag-of-bones model from BANMo [65], which accounts for both articulated motion and non-rigid deformation. The deformation is computed by blending the motion of a set of unstructured 3D coordinates/bones that rotate and translate over time.
>
> **Q2 In particular, it is not clear how eq.2 is combined with eq.3, and why the same number of "bones" (b=25, L.137) is used for all agents. Also, the nature of G^b is not discussed in detail.**
>
> We model the density and color of an agent in the time-invariant space (Eq. 2), which can be mapped to the deformed space at a given time instance (Eq. 3). We use $B=25$, as it is the superset for bones of all agents we processed. $G^b_t$ is a rigid transformation representing the state of bone b at time t.
>
> **Q3 How NeRF-type reconstructions are combined with feature descriptors, and how this helps in handling layout changes?**
>
> Similar to distilled feature fields [A], we extend a static NeRF to represent feature fields (Eq. 1), and optimize them together with the motion $\mathcal{D}$ using DINOv2 descriptors. This is referred to as feature-metric bundle adjustment (FBA) in Eq. 7. We find FBA is robust to moderate layout changes, since DINOv2 feature descriptors are robust to local appearance changes [40]. We quantitatively validate the effect of FBA in [global response](https://openreview.net/forum?id=fzdFPqkAHD&noteId=abfRu0bgF7) Tab B. *This is also evident in Fig. D of the rebuttal pdf, where we can localize cameras despite of the layout changes of the scene.*
>
> [A] Kobayashi, Sosuke, Eiichi Matsumoto, and Vincent Sitzmann. "Decomposing nerf for editing via feature field distillation." NeurIPS 2022.
>
> **Q4 Clarification on designs (Eq. 6-7, Eq. 10-11).**
>
> We added an ablation of Eq. 10-11 in Table D row (d). We find that replacing ControlUNet with concatenation (L197-198, concatenating goals with perception codes) produces worse results (e.g., Path error: 0.115 vs 0.146). We also provided additional quantitative comparisons and analysis on camera localization (Eq. 6), feature-metric bundle adjustment (Eq. 7) in the [global response](https://openreview.net/forum?id=fzdFPqkAHD&noteId=abfRu0bgF7) Table B. We found that those designs are necessary to achieve good performance.
>
> Table D: Evaluation of Behavior Control. We separately evaluate path and full body motion generation, given guidance signals of goal and path respectively. The metrics are minimum average displacement error (minADE) with standard deviations (±σ). The best results are in bold.
> | Method                         | Path (m) ↓        | Orientation (rad) ↓  | Joint Angles (rad) ↓ |
> |-------------------------------|-------------------|----------------------|----------------------|
> | Gaussian [31, 44]             | 0.206±0.002       | 0.370±0.003          | 0.232±0.001          |
> | ATS (Ours)                    | **0.115±0.006**   | **0.331±0.004**      | **0.213±0.001**      |
> |                               |                   |                      |                      |
> | (a) w/o observer $ω_o$        | 0.126±0.011       | **0.330±0.004**          | **0.212±0.001**          |
> | (b) w/o scene $ω_s$           | 0.179±0.003       | **0.329±0.004**          | **0.212±0.001**          |
> | (c) ego→world [61]            | 0.209±0.002       | 0.429±0.006          | 0.250±0.002          |
> | (d) control-unet→concat         | 0.146±0.005       | 0.351±0.004          | 0.220±0.001          |
>
> **Q5 What happens if a behavior is observed only once in the dataset or conversely, how many times does a behavior need to be observed to be included in the model?**
>
> Due to our ego-centric encoding (Eq.12), we find that a behavior can be learned and generalized to novel situations even when seen once. Although there's only one data point where the cat jumps off the dining table, our method can generate diverse motion of cat jumping off the table while landing at different locations (to the left, middle, and right of the table). *Please see Fig B of the [rebuttal pdf](https://openreview.net/attachment?id=abfRu0bgF7&name=pdf) for the corresponding visual.*
>
> **Q6 How robust is the method with respect to changes in the environment?**
>
> We noticed displacements of chairs and the presence of new furniture in our captured data. Our method is robust to these in terms of camera localization (Tab B of the global response, Fig D of the rebuttal pdf). However, 3D reconstruction of these transient objects is challenging and we leave it as future work.
>
> **Q7 What "longitudinal videos" are?**
>
> Longitudinal videos come from the term “longitudinal study”, which refers to a research design that involves repeated observations of the same variables (e.g., people) over long periods of time. We think it fits well with our study of learning a behavior model of agents from videos captured over a long-horizon. We will clarify.
>
> **Q8 What does "low latency" mean in this context?**
>
> Low-latency indicates the model can generate goals at an interactive frame-rate. We will clarify.

---

> ### Comment · Reviewer_m6Ge · 2024-08-13
> **Comments after rebuttal**
>
> I thank the authors for their responses and their comments. I appreciate the additional results provided as well as the evaluation regarding camera localization and 4D reconstruction. Their answers have clarified some aspects, especially regarding agent reconstruction. As mentioned in the reviews, there are issues regarding notation and clarity which, to some extent, are due to the limited space available for describing in sufficient details all the claimed contributions. In my view, the one additional page of the final version will still not be sufficient. This also concerns reproducibility, as sufficient details in the paper itself need to be provided, even if the code will be published. I increase my rating to borderline reject, as I appreciate the contributions of this work and the answers provided by the authors, yet I feel that a more detailed presentation and evaluation would make the paper much stronger.

---

### Author Rebuttal · Authors · 2024-08-06

We would like to thank the reviewers for their feedback. We propose an approach to learn an interactive behavior model of agents from casual videos captured over a long time horizon. Reviewers note that we tackle a "challenging" problem (m6Ge, Kj9h) with “very interesting”, “effective” ideas (m6Ge, C9Gr), “great technical achievement to reconstruct agent behavior” (xDUi) as well as “great visuals” (Kj9h) that “may inspire researchers in the field of 4D reconstruction” (C9Gr).

This paper received 3 above-accept reviews: Accept, WA, BA, and one Reject. The reject recommendation is due to the lack of space and detailed description given the complexity of the problem addressed. Other reviewers request quantitative evaluations. We report more quantitative results on behavior prediction (m6Ge, Kj9h), camera registration and 4D reconstruction (m6Ge, Kj9h, C9Gr). Please also note that we will make our code/data available for reproducibility, and improve the exposition based on the feedback using the extra 1 page allowance.

**End-to-end evaluation and comparison to a 1-stage model (Kj9h)**: We re-did the evaluation of the behavior prediction using the suggested end-to-end setup without using GT goals/paths (please see the new Table A), and added comparison against the 1-stage model (row a). Our hierarchical model out-performs 1-stage by a large margin for all metrics. We posit hierarchical model makes it easier to learn individual modules. We also re-run the other ablations in this setting (row b-d), which verifies our design choices.

Table A: End-to-end Evaluation of Interactive Behavior Prediction. We report results of predicting goal, path, orientation, and joint angles, using $K = 16$ samples across $L = 12$ trials. The metrics are minimum average displacement error (minADE) with standard deviations (±σ). The best results are in bold.

| Method                          | Goal (m) ↓        | Path (m) ↓           | Orientation (rad) ↓  | Joint Angles (rad) ↓ |
|---------------------------------|-------------------|----------------------|----------------------|----------------------|
| Location prior [94]             | 0.663±0.307       | N.A.                 | N.A.                 | N.A.                 |
| Gaussian [31, 44]               | 0.942±0.081       | 0.440±0.002          | 1.099±0.003      | 0.295±0.001      |
| ATS (Ours)                      | **0.448**±0.146   | **0.234**±0.054      | **0.550**±0.112      | **0.237**±0.006      |
|                                 |                   |                      |                      |                      |
| (a) hier→1-stage [73]           | 1.322±0.071       | 0.575±0.026          | 0.879±0.041          | 0.263±0.007          |
| (b) w/o observer $ω_o$          | 0.647±0.148       | 0.327±0.076          | 0.620±0.092          | 0.240±0.006          |
| (c) w/o scene $ω_s$             | 0.784±0.126       | 0.340±0.051          | 0.678±0.081          | 0.243±0.007          |
| (d) ego→world [61]              | 1.164±0.043       | 0.577±0.022          | 0.873±0.027          | 0.295±0.006      |

**Camera localization (m6Ge, Kj9h, C9Gr)**: We added an experiment on camera localization using GT cameras from annotated GT correspondences. *A visual of the annotated GT correspondence and 3D alignment can be found in Fig. C of the attached pdf.*

We report camera translation and rotation errors in Table B. We observe that removing neural localization (Eq. 6) produces significantly larger localization error (e.g., Rotation error: 6.35 vs 37.56). Removing feature-metric bundle adjustment (Eq. 7) also increases the error (e.g., Rotation error: 6.35 vs 22.47). Our method outperforms multi-video TotalRecon by a large margin due to the above innovations.

Table B: Evaluation of Camera Registration: The best results are in bold.
| Method                      | Rotation Error (°) ↓ | Translation Error (m) ↓ |
|-----------------------------|----------------------|-------------------------|
| Ours                        | **6.35**             | **0.41**                |
| w/o Neural Localizer        | 37.59                | 0.83                    |
| w/o Featuremetric BA        | 22.47                | 1.30                    |
| Multi-video TotalRecon      | 59.19                | 0.68                    |

**4D reconstruction (m6Ge, Kj9h, C9Gr)**. We added an experiment to evaluate the accuracy of 4D reconstruction using synchronized videos captured with two moving iPhone cameras looking from opposite views. We compute the GT relative camera pose between the two cameras from 2D correspondence annotations. One of the synchronized videos is used for 4D reconstruction, and the other one is used as held-out test data. For evaluation, we render novel views from the held-out cameras and compute novel view depth accuracy DepthAcc (depth accuracy thresholded at 0.1m) for all pixels, agent, and scene, following TotalRecon [52].

Our method outperforms both the multi-video and single-video versions of TotalRecon by a large margin in terms of depth accuracy and LPIPS, due to the ability of leveraging multiple videos. *Please see Fig A in the rebuttal pdf for qualitative comparison.*

Table C: Evaluation of 4D Reconstruction. The best results are in bold.
| Method         | DepthAcc (all) ↑ | DepthAcc (fg) ↑ | DepthAcc (bg) ↑ | LPIPS (all) ↓ | LPIPS (fg) ↓  | LPIPS (bg) ↓  |
|----------------|------------------|-----------------|-----------------|---------------|---------------|---------------|
| Ours           | **0.708**        | **0.695**       | **0.703**       | **0.613**     | **0.609**     | **0.613**     |
| Single-video TotalRecon     | 0.533            | 0.685           | 0.518           | 0.641         | 0.619         | 0.641         |
| Multi-video TotalRecon     | 0.093            | 0.644           | 0.047           | 0.622         | 0.616         | 0.623         |

---

### Decision · Program_Chairs · 2024-09-25

**Decision:**

Reject

**Comment:**

Claimed contributions (according to the intro)
 1. Agent-to-sim framework to learn the behavior of an agent in a specific environment (e.g. a cat at home) from multiple, casually-captured videos for a long time (e.g. a month)
 2. Environment-interactive simulation (both to the observer and the 3D scene)

Strengths
 - Challenging end-to-end task of 4D reconstruction of 3D scene and agent motions from videos to obtain data to train an environment-aware agent
 - Behavior generation with a motion diffusion model conditioned by eco-centric perception

Weaknesses
 - The paper is too densely packed with missing details. Even the subproblems here, 4D reconstruction, deformable body tracking, and environment-aware agent learning are challenging enough to be papers by themselves
 - Reproducibility
 - Insufficient evaluations, especially on the environment-aware agent simulation, which is supposed to be the goal of the project

The paper would have had a stronger alignment with the claims if it spent more space on the agent learning and its evaluations, not the data pipeline mostly relying on off-the-shelf technology and techniques. Therefore, I recommend rejecting the paper.